# SMQVP: A Web Application for Spatial Metabolomics Quality Visualization and Processing

**DOI:** 10.3390/metabo15060354

**Published:** 2025-05-27

**Authors:** Zhanlong Mei, Wan Sun, Yun Zhao, Haoke Deng, Xiaolian Ning, Chunlu Feng, Jin Zi

**Affiliations:** 1BGI Genomics, Shenzhen 518083, China; meizhanlong@genomics.cn (Z.M.); sunwan@genomics.cn (W.S.); zhaoyun@genomics.cn (Y.Z.); denghaoke@genomics.cn (H.D.); fengchunlu@genomics.cn (C.F.); 2BGI Research, Shenzhen 518083, China; ningxiaolian@genomics.cn

**Keywords:** spatial metabolomics, mass spectrometry imaging, data quality control, noise ion filtering, isotopic peak detection

## Abstract

Background: Spatial metabolomics is a powerful technique that enables spatially resolved mapping of metabolite distributions at the tissue and cellular levels, providing valuable insights into biological processes. However, challenges in data quality control and preprocessing remain significant bottlenecks, critically impacting the reliability of downstream analyses and the robustness of findings. Methods: To address these limitations, we present Spatial Metabolomics data Quality Visualization and Processing (SMQVP v1.0), a novel software with a user-friendly graphical interface designed for the systematic quality assessment and preprocessing of spatial metabolomics data. SMQVP incorporates eight comprehensive quality visualization and evaluation modules, including background consistency assessments, noise ion filtering, intensity distribution analyses, and the identification of isotopic and adduct ions. Results: We demonstrated SMQVP’s effectiveness using AFADESI-based mouse brain data, showing that the tool successfully identified and removed noise signals. This rigorous preprocessing resulted in improved clustering outcomes that more accurately reflected the underlying tissue morphology compared with analyses performed on unprocessed data. Conclusions: SMQVP is the first systematic approach focused on quality visualization, specifically for spatial metabolomics. It offers researchers an accessible and comprehensive solution for enhancing data integrity and mitigating the impact of technical noise, thereby improving the reliability and robustness of their spatial metabolomics findings.

## 1. Introduction

Metabolomics is a rapidly advancing field of considerable significance in modern biology and biomedical research [1,2,3]. Among the various metabolomic approaches, spatial metabolomics is particularly powerful. By enabling the mapping of metabolite distributions with spatial resolution at the tissue and even cellular levels, it offers powerful capabilities for elucidating biological processes and disease mechanisms. This field is supported by several key technological platforms, including Matrix-Assisted Laser Desorption/Ionization Mass Spectrometry Imaging (MALDI-MSI) [4,5], Desorption Electrospray Ionization Mass Spectrometry Imaging (DESI-MSI) [6], Air Flow-Assisted Desorption Electrospray Ionization (AFADESI) [7], and Secondary Ion Mass Spectrometry (SIMS) [8]. However, all mass spectrometry imaging techniques have inherent challenges [9,10].

Regardless of the specific spatial metabolomics platform used, generating high-quality, reliable data is crucial for accurate analyses and meaningful biological insights. Common challenges encountered include overlapping mass spectrometry signals, high levels of chemical noise, signal intensity instability across the sample, and difficulties in achieving accurate quantification [11]. These issues substantially complicate data processing and can reduce the reliability of downstream biological interpretations. Although several software tools exist for spatial metabolomics data analyses [12]—for example, commercial options such as MassImager [13] and MsiReader [14], and non-commercial packages such as Cardinal [15], multi-MSIProcessor [16] and mzMINE3 [17]—they often lack comprehensive modules dedicated specifically to data quality assessments and systematic preprocessing. Consequently, researchers frequently need to rely on considerable expertise to evaluate the quality of data and to perform the necessary preprocessing steps, posing a significant technical barrier, particularly for those new to the field.

To tackle the critical challenges in spatial metabolomics data analysis, we developed SMQVP (Spatial Metabolomics data Quality Visualization and Processing), an innovative software with a user-friendly graphical interface tailored to integrated quality visualization and preprocessing. SMQVP offers a robust suite of tools, encompassing eight quality visualization points (QVPs) and three core preprocessing functions, including background pixel removal, noise ion filtering, and the identification of isotopic and adduct ions. Its visualization modules enable comprehensive assessments, such as evaluating specific ion spatial distributions, analyzing background region consistency, examining overall intensity and missing value distributions, and assessing isotopic peak and adduct ion ratios (Appendix A). SMQVP is platform-agnostic and compatible with data from diverse spatial metabolomics technologies. In this study, we have validated SMQVP’s performance using both an AFAD-ESI-based dataset and MALDI-MSI datasets. A comparison of SMQVP’s key features and capabilities relative to existing spatial metabolomics software is provided in Appendix A. By integrating these capabilities, SMQVP provides researchers with a streamlined, accessible solution for data quality control and preprocessing, significantly reducing technical barriers and improving the reliability of spatial metabolomics studies.

## 2. Materials and Methods

### 2.1. Workflow of SMQVP

The overall workflow implemented by SMQVP is depicted in Figure 1A. The process begins with Data Upload, followed by initial Spatial Visualization. This visualization step facilitates the examination of characteristic ion distributions and enables the user to delineate distinct tissue and background regions using interactive tools. Following region delineation, the workflow proceeds to Background and Tissue Quality Assessment. This stage involves inspecting and analyzing the spectral profiles and consistency within the selected background regions. Additionally, the ion intensity levels are compared between the tissue and background regions to identify ions significantly enriched in the tissue. Based on these assessments, two key preprocessing steps are performed: Background Pixel Removal and Noise Ion Filtering. SMQVP removes pixels classified as part of the background from the dataset using a threshold-based method, as detailed in Section 2.2.3. Subsequently, noise ions, characterized by random spatial distributions [18], are identified and filtered out. After these initial filtering steps, SMQVP provides modules for Overall Data Characteristic Visualization. This includes analyzing and displaying the distribution of total intensity and missing values across both pixels and ions. The final stage of the workflow involves Specific Peak Identification, focusing on identifying potential isotopic peaks and adduct ions within the processed dataset. The final processed files, including the peak intensity table and the peak and pixel information table, can then be downloaded.

### 2.2. Introduction to Quality Visualization and Processing Points

#### 2.2.1. QVP1 and Process 1: Spectral Analysis of Background and Tissue Regions

Upon data upload, SMQVP generates a plot displaying the total intensity distribution across all pixels. Process 1 involves users employing an interactive lasso tool within the graphical interface to manually delineate the tissue and background regions. For each defined region, the median intensity of each ion across its pixels is calculated to create a representative spectrum. QVP1 focuses on a critical quality assessment by visually comparing the spectra of different background regions, which should exhibit high similarity, with the tissue region spectrum, which should be distinctly different. Additionally, this module enables users to inspect spectra for polymer peaks or other artifacts, enhancing the reliability of the data quality evaluation.

#### 2.2.2. QVP2: Background Region Consistency

Spatial metabolomics data acquisition can span significant durations. The consistency observed across different selected background regions provides an indicator of instrument stability during the acquisition period. QVP2 quantifies this consistency by calculating the Pearson correlation coefficient of ion peak intensities between the spectra of different background regions. A correlation coefficient [19] typically exceeding 0.9 suggests a high degree of consistency and indicates relatively stable data acquisition.

#### 2.2.3. QVP3 and Process 2: Distinction and Proportion of Tissue-Enriched Ions

QVP3 involves analyzing ion expression to identify tissue-enriched ions. Using the user-defined tissue and background regions, pixels are assigned to their respective groups. For each ion, a Wilcoxon rank-sum test is conducted to compare the intensities between the tissue and background groups, calculating the fold change and *p*-value. Ions with a fold change greater than 1 and a *p*-value less than 0.05 are classified as tissue-enriched, and the proportion of such ions is determined to assess the data quality.

Process 2 utilizes these tissue-enriched ions to generate a total intensity distribution plot for all pixels. Users can set an adjustable intensity threshold to classify pixels as background (below the threshold, removed) or tissue (above the threshold, retained). This threshold-based segmentation refines the spatial delineation, effectively excluding background pixels from downstream analyses while preserving relevant ions for further investigation.

#### 2.2.4. QVP4 and Process 3: Identification and Proportion of Noise Ions

Noise ions, characterized by their random spatial distribution, can mask biologically significant spatial patterns in imaging mass spectrometry data. To address this, the SMQVP software utilizes the quadrat test from the spatstat package [20], a statistical test for Complete Spatial Randomness (CSR), to identify ions with statistically random spatial distributions. For each ion, a “noise score” is calculated as the negative base-10 logarithm of the quadrat test’s *p*-value. Ions with noise scores below a user-defined threshold are classified as potential noise ions. The proportion of ions identified as noise, expressed as a percentage, is referred to as QVP4. These noise ions are subsequently removed from the dataset in a step termed Process 3. SMQVP provides empirically derived default thresholds (e.g., noise score = 60) but encourages users to iteratively refine these values using interactive visualizations. To aid in threshold selection, SMQVP generates spatial distribution maps for ions near the threshold. This functionality allows users to visually assess whether these borderline ions exhibit a random spatial distribution, thereby facilitating precise optimization of the noise score threshold based on their dataset’s characteristics.

#### 2.2.5. QVP5: Analysis of Intensity Distribution

The median intensity for both individual pixels and ions is calculated and visualized as QVP5. For pixels, a spatial distribution map depicting the median intensity across the sample is generated. For ions, a spectrum distribution plot (plotting median intensity vs. *m*/*z*) and a frequency distribution histogram of median ion intensities are provided to assess the overall signal strength and distribution patterns.

#### 2.2.6. QVP6: Assessment of Missing Value Distribution

QVP6 evaluates the distribution of missing values by calculating two metrics: (1) the missing pixel ratio, defined as the percentage of ions undetected in each pixel, and (2) the missing ion ratio, defined as the percentage of pixels where a specific ion was undetected. Spatial visualization of these ratios identifies regions or ions with poor data coverage.

#### 2.2.7. QVP7: Isotopic Peak Ratio Assessment

QVP7 quantifies the proportion of isotopic ions, serving as a crucial metric for assessing mass spectrometry data quality. Isotopic peaks are expected features of most endogenous metabolites in mass spectrometry data. While low-intensity isotopic peaks may not always be detected, their presence in assessments of high-abundance ions is a positive indicator of reliable detection. SMQVP utilizes the isotopologues function from MetaboCoreUtils [21] to preliminarily identify potential isotopic peak pairs. This involves searching for peak pairs in which the measured mass difference deviates from the theoretical isotopic mass difference (e.g., ~1.0035 Da for the M + 1 isotope) by no more than a user-specified mass error tolerance (ppm). For these putative pairs, the Pearson correlation of their spatial intensities across pixels is calculated. Pairs exhibiting a spatial correlation exceeding a specified user-defined threshold are then confidently assigned as true isotopic peaks. Evaluating the presence and expected proportions of these isotopic peaks, especially for high-intensity ions, is a valuable approach to evaluating overall data quality and confirming the reliability of ion detection.

#### 2.2.8. QVP8: Adduct Ion Ratio Assessment

QVP8 represents the proportion of identified adduct ions. The accurate identification of different adduct forms of metabolites is essential for correct compound annotation and downstream analysis. SMQVP facilitates this by utilizing predefined lists of common adduct mass differences for both the positive and negative ion modes, which are also customizable by the user. The software systematically searches the dataset for pairs of ions whose mass difference corresponds to these specified adduct mass differences. For each potential adduct pair identified, the Pearson correlation of their spatial intensities across the dataset is calculated. Pairs exhibiting a spatial correlation exceeding a user-defined threshold are then confidently assigned as potential adduct ions. This module is critical for verifying the consistent presence of potential adduct forms across the spatial dimension, thereby enhancing confidence in the analytical results.

### 2.3. Implementation of SMQVP

SMQVP (Spatial Metabolomics Quality Visualization and Processing) is a user-friendly graphical user interface tool designed to facilitate quality control and preprocessing of spatial metabolomics data. Built using the Shiny package [22] in R, SMQVP offers an intuitive platform that enables researchers to interactively assess and enhance data quality through visualizations and analyses. The tool is accessible online at https://metax.genomics.cn/app/SMQVP (accessed on 22 May 2025), hosted on a cloud server equipped with 128 CPU cores and 1000 GB of RAM, featuring a clear layout with a navigation panel on the left and an analysis content panel on the right (Figure 1B). The navigation panel provides access to key modules, such as Help, Data Upload, and quality control functions, allowing users to upload their own datasets or explore the tool using built-in demo data. Details on SMQVP’s computational performance, including its execution time and memory usage on various dataset sizes, are provided in Appendix A. For users preferring local deployment, the source code of SMQVP v1.0 is available on GitHub at https://github.com/mzlab-research/SMQVP.git (accessed on 22 May 2025). SMQVP requires input data in the form of a peak table matrix, where the first two columns represent the X and Y spatial coordinates of each pixel, followed by columns for different *m*/*z* values (ions), with each cell indicating the ion intensity at that pixel. This peak table format makes SMQVP compatible with data generated from various spatial mass spectrometry imaging platforms, such as MALDI-MSI, DESI-MSI, and AFADESI, upon conversion of raw data to this format. Detailed guidance on formatting this input is available in the tool’s tutorial panel. Upon completion of the quality control and preprocessing workflow, SMQVP provides users with downloadable output files. These files include the processed peak intensity table, along with comprehensive tables containing detailed information for each peak (including ion intensity, missing ion ratio, isotope and adduct status, and noise score) and for each pixel (including pixel intensity and missing pixel ratio).

### 2.4. Demo Data Acquisition

To demonstrate the capabilities of SMQVP, we prepared two independent experimental datasets. The first demonstration dataset was acquired from 7-week-old male mouse brain tissue using the AFADESI spatial metabolomics platform in the positive ion mode. The data acquisition parameters included a spray solvent composition of acetonitrile and water (80:20 *v*/*v*), an AFADESI extraction gas flow rate of 45 L/min, and a nebulizing gas (nitrogen) flow rate of 0.65 MPa. The spatial resolution of the acquisition was 50 μm, and the mass spectrometry resolution was 70,000. The raw data were processed using the Cardinal package [15] to generate the peak table matrix. The final dataset comprised 53,812 pixels and 2654 ion peaks.

To further establish SMQVP’s cross-platform compatibility, a second demonstrative dataset was acquired from adult mouse brain tissue. This acquisition employed an atmospheric pressure scanning microprobe matrix-assisted laser desorption/ionization (AP-SMALDI) imaging platform (TransMIT GmbH) interfaced with a Q-Exactive Orbitrap mass spectrometer (Thermo Fisher Scientific, Waltham, MA, USA). Data were collected in the negative ion mode, with a spatial resolution of 20 µm and a mass spectrometry resolution of 140,000. Subsequently, the raw mass spectrometry imaging data underwent identical preprocessing using the Cardinal package to generate comparable peak intensity matrices. Representative analytical results derived from SMQVP’s processing of this AP-SMALDI dataset are depicted in Appendix A, thereby underscoring its consistent performance across diverse technological platforms.

## 3. Results and Discussion

We evaluated the performance and utility of SMQVP using a spatial metabolomics dataset acquired from mouse brain tissue via the AFADESI platform. A corresponding Hematoxylin and Eosin (H&E)-stained image of the mouse brain section is shown in Figure 2A.

Following data upload, the initial exploration involved examining the spatial distribution of characteristic ions. Choline (*m*/*z* 104.1069), a metabolite known to exist in animal tissue, was selected as an example. Its spatial distribution exhibited high intensity throughout the tissue section, with minimal signal detected in the background areas (Appendix A). Based on the median pixel intensity visualization, we manually delineated five distinct tissue and background regions within the SMQVP interface (Appendix A). Applying QVP1, we plotted the median ion intensity spectra for these regions (Figure 2B). As expected, the spectral profiles of the background regions showed high similarity to each other and were distinctly different from the tissue region spectrum. No polymer peaks were detected in either region. QVP2 quantified the spectral consistency across the selected background regions by calculating Pearson correlation coefficients (Figure 2C). All selected background areas exhibited high correlations, consistently above 0.99, indicating generally stable signal acquisition. However, hierarchical clustering of the background spectra revealed three sub-clusters (regions 1 and 2; regions 3 and 5; and region 4), suggesting minor systematic variations in background signals during acquisition.

QVP3 analyzed the proportion of ions with significantly higher expression in the tissue regions compared with that in the background (Figure 2D). This analysis showed that 37% of the detected ions were enriched in the tissue. The proportion of tissue-enriched ions varies across experimental platforms (e.g., MALDI vs. DESI) and tissue types. While no universal threshold exists, higher proportions generally indicate superior data quality by reflecting stronger retention of tissue-specific signals. SMQVP utilizes these tissue-enriched ions to construct a total intensity image of the sample (Appendix A). By applying a total intensity threshold of 107.37, background pixels could be clearly distinguished and effectively removed from the dataset (Appendix A).

After removing the background pixels, QVP4 focused on identifying noise ions, defined as ions with random spatial distributions. The overall distribution of noise scores across all ions is shown in Figure 2E. We examined the spatial distribution maps for ions with representative noise scores near 20, 60, and 90 (Appendix A). Ions with scores around 20 showed no discernible spatial structure, consistent with noise. Ions with scores of 60 began to show some spatial features, while those with scores of 90 exhibited clear, non-random spatial patterns. Based on these visualizations, we selected a noise score threshold of 90 to filter out randomly distributed ions, retaining 37% of the original ions for further analysis.

QVP5 provided insights into the overall data characteristics by visualizing the spatial distribution of total pixel intensity (Figure 2F) and the frequency distribution of ion intensities (Figure 2G). The total pixel intensity reached approximately 1 × 10^7^, and the ion intensity distribution approximated a normal distribution. QVP6 assessed the distribution of missing values at both the pixel and ion levels (Figure 2H,I). The majority of pixels had missing values below 5%, with slightly higher proportions observed only at the tissue edges. Similarly, most ions exhibited missing values below 5% across the dataset.

QVP7 focused on identifying isotopic peaks among the remaining ions (Figure 2J). High-intensity ions reliably showed matching isotopic peaks, confirming the detection quality. Overall, SMQVP identified isotopic peaks in 14.44% of the ions. Finally, QVP8 identified potential adduct ions after removing non-monoisotopic peaks. Using a predefined list of seven common adduct types, we identified adducts for 15.88% of the filtered ions. M + H was the most prevalent adduct, followed by significant proportions of M + K and M + Na, consistent with prior observations in datasets of similar tissue [23].

Through these comprehensive quality assessment modules, we gained a thorough understanding of the data characteristics, including background stability, noise levels, signal intensity distributions, missing data patterns, and the presence of isotopic and adduct ions.

To demonstrate the impact of SMQVP’s preprocessing steps on a downstream analysis, we performed Louvain clustering [24,25] on the data both before and after processing. The clustering results for the preprocessed data (Figure 3A) showed that background areas formed distinct clusters, consistent with the minor systematic variations identified in QVP2. Crucially, the tissue region clustering failed to accurately reflect known neuroanatomical structures, likely due to interference from background noise and randomly distributed ions. In contrast, clustering performed on the data after preprocessing with SMQVP (Figure 3B) yielded distinct spatial regions that closely resembled the morphology observed in the H&E staining (Figure 2A). This marked improvement in spatial clustering demonstrates that SMQVP’s preprocessing effectively removes confounding signals, resulting in higher-quality data more suitable for biological interpretation and spatial pattern analysis.

## 4. Conclusions

While spatial metabolomics holds great potential, challenges in data quality control and preprocessing limit its reliability. We address this need with SMQVP, the first dedicated, user-friendly software integrating comprehensive quality visualization and essential preprocessing for spatial metabolomics data. SMQVP’s systematic modules enable effective data cleaning, leading to improved integrity and more reliable downstream analysis, as demonstrated through enhanced spatial clustering. By providing an accessible, standardized platform, SMQVP lowers technical barriers and accelerates discovery in spatial metabolomics.

## Figures and Tables

**Figure 1 metabolites-15-00354-f001:**
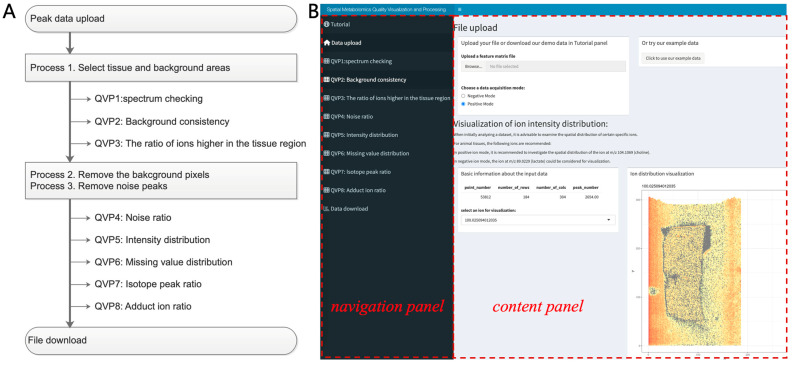
The workflow (**A**) and a screenshot (**B**) of SMQVP.

**Figure 2 metabolites-15-00354-f002:**
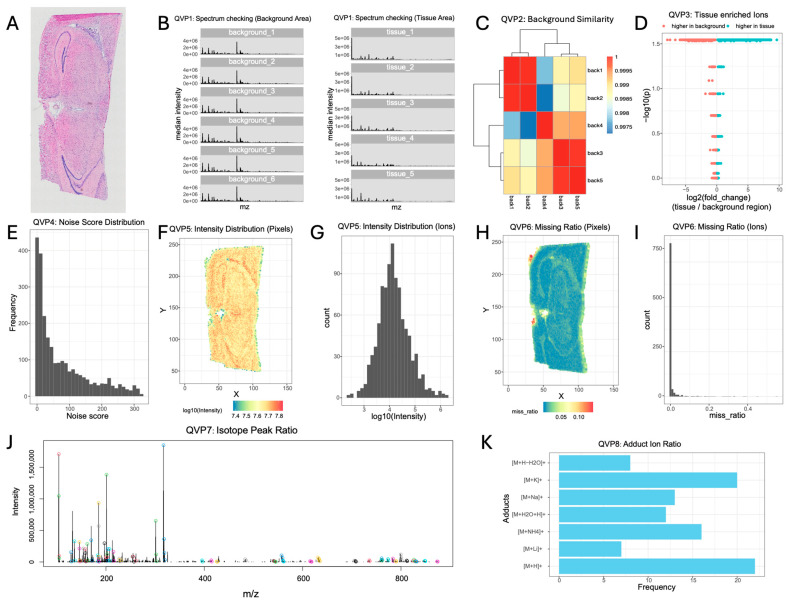
Quality visualization results of SMQVP. Hematoxylin and Eosin (H&E)-stained image (**A**); QVP1: mass spectral analysis of background and tissue areas (**B**); QVP2: background similarity shown through a correlation heatmap of mass spectra from the background areas (**C**); QVP3: proportion of tissue-enriched ions (**D**); QVP4: distribution of noise scores (**E**); QVP5: intensity distribution of pixels (**F**) and ions (**G**); QVP6: missing value distribution of pixels (**H**) and ions (**I**); QVP7: distribution of isotopic peaks (**J**); and QVP8: distribution of adduct ions (**K**).

**Figure 3 metabolites-15-00354-f003:**
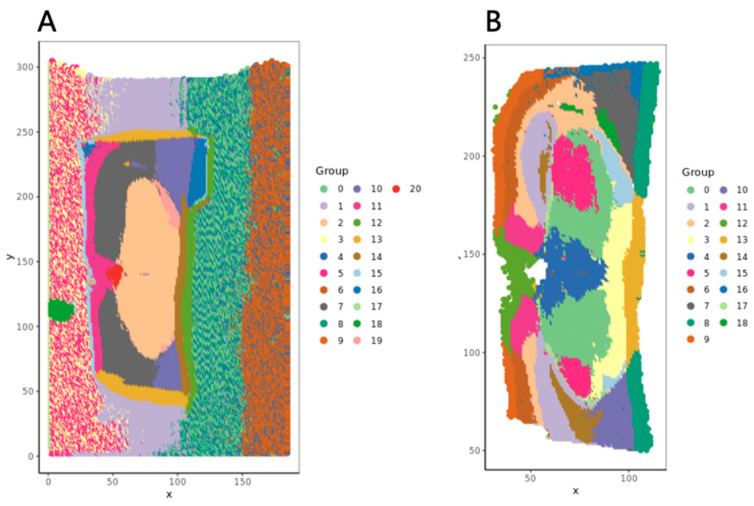
The Lovain clustering results for mouse brain data before (**A**) and after (**B**) data preprocessing.

## Data Availability

All resources described in this study are publicly available. The raw spatial metabolomics data and processed peak intensity tables derived from the mouse brain tissue have been deposited in the OMIX database of the National Genomics Data Center. The AFADESI dataset is available under accession number OMIX009541 (https://ngdc.cncb.ac.cn/omix/release/OMIX009541, accessed on 22 May 2025), while the AP-SMALDI dataset is accessible via accession number OMIX010192 (https://ngdc.cncb.ac.cn/omix/release/OMIX010192, accessed on 22 May 2025).

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
