# Peer review of "SMQVP: A Web Application for Spatial Metabolomics Quality Visualization and Processing"

_metabolites, 2025, doi:10.3390/metabo15060354_

Round 1

Reviewer 1 Report

Comments and Suggestions for Authors
  • In the title “SMQVP: a web application for spatial metabolomics quality visulization and processing”, visulization should be visualization. Suggest spelling and grammar check including capital letters in the whole manuscript.
  • Page 2 line 60, “By integrating these capabilities, SMQVP provides researchers with a streamlined, accessible solution for data quality control and preprocessing, significantly reducing technical barriers and improving the reliability and reproducibility of spatial metabolomics studies.” How does this software improve the reproducibility?
  • Figure 1B is not clear.
  • Page 2 line 74, “Pixels classified as background are removed from the dataset.” did not state how background was defined.
  • Page 3 line 97, “the Pearson correlation coefficient” should provide a reference.
  • Page 4 line 144, “a specified mass error tolerance (ppm)”, what is the ppm value that was used?
  • Page 4 line 168, “The tool is accessible online at https://metax.genomics.cn/app/SMQVP, featuring a clear layout with a navigation panel on the left and an analysis content panel on the right (Figure 1B).”, I did not see the analysis content panel on the right on the website.
  • Page line 179, “These files include the processed peak intensity table, along with comprehensive tables containing detailed information for each peak (including intensity, missing ratio, isotope and adduct status, and noise score) and for each pixel (including intensity, missing ratio).”, missing ratio was not defined.
  • Page 6 line 247, reference “Louvain clustering[17]” did not use this term. Need clarify.
  • Figure 2 is not clear. Labels are not readable, so I can’t review Figure 2.

Author Response

1. Summary

We sincerely thank you for the rigorous critique, which has strengthened SMQVP’s methodological transparency and usability. The revised manuscript addresses technical ambiguities by clarifying background pixel classification (Section 2.2.3), defining "missing ratio" metrics (Section 2.2.6). Critical figures were optimized for clarity—enlarging labels in Figure 1B, restructuring Figure 2 with explicit QVP annotations, and refining captions. Citations (Louvain clustering, Pearson correlation) were standardized with updated references. Claims about reproducibility were refined to focus on workflow standardization. These revisions enhance methodological rigor, ensure visual clarity, and improve user guidance across SMQVP’s quality control modules.

2. Point-by-point response to Comments and Suggestions for Authors

Comment 1: "In the title 'SMQVP: a web application for spatial metabolomics quality visulization and processing', visulization should be visualization. Suggest spelling and grammar check including capital letters in the whole manuscript."

Response: Thank you for highlighting the typographical error in the manuscript title. We sincerely apologize for this oversight and have revised the title to: "SMQVP: A Web Application for Spatial Metabolomics Quality Visualization and Processing" (Page 1, Line 2).

We have performed a thorough spelling and grammar check throughout the manuscript, ensuring proper capitalization (e.g., "Web Application" in the title).  

Comment 2: "Page 2 line 60, 'By integrating these capabilities, SMQVP provides researchers with a streamlined, accessible solution for data quality control and preprocessing, significantly reducing technical barriers and improving the reliability and reproducibility of spatial metabolomics studies.' How does this software improve the reproducibility?" 

Response:  Thank you for your thoughtful comment. We agree that the original statement might have overstated the software’s impact on reproducibility, which involves multiple factors beyond data preprocessing. Our intended meaning was that SMQVP promotes more consistent data handling through a standardized quality control workflow, thereby supporting reliability in downstream analyses.

To prevent ambiguity, we have revised the sentence to focus on workflow streamlining and improved reliability, without directly referencing reproducibility. The revised description: "By integrating these capabilities, SMQVP provides researchers with a streamlined, accessible solution for data quality control and preprocessing, significantly reducing technical barriers and improving the reliability of spatial metabolomics studies. (Page 2, Line 69)"

Comment 3: "Figure 1B is not clear." 

Response: Thank you for your comment. We have revised Figure 1B by capturing a new screenshot with significantly enlarged font sizes for improved readability. We also verified all labels and text in the figure are legible at both on-screen and print resolutions.

Comment 4: "Page 2 line 74, 'Pixels classified as background are removed from the dataset.' did not state how background was defined." 

Response: We appreciate your comment regarding the definition of background pixels.

This sentence appears within Section 2.1, which outlines the overall workflow of SMQVP at a high level. The detailed methodology for how pixels is classified as background and subsequently removed is described in Section 2.2.3 ("QVP3 and process 2: Distinction and Proportion of Tissue-Enriched Ions").

As explained in Section 2.2.3, after initial user-defined region delineation, tissue-enriched ions are identified. Process 2 then utilizes these ions to calculate a total intensity distribution for all pixels, allowing users to set an adjustable intensity threshold to classify pixels as background (below threshold) or tissue (above threshold) for removal.

To enhance clarity for the reader navigating the workflow, we have revised the sentence to include a reference to Section 2.2.3 as: " SMQVP removes pixels classified as background from the dataset using a threshold-based method as detailed in Section 2.2.3. (Page 2, Line 84)"

Comment 5: "Page 3 line 97, 'the Pearson correlation coefficient' should provide a reference." 

Response: Thank you for this comment. We agree that adding a reference for the Pearson correlation coefficient on Page 3, line 97 is appropriate. We have added the following reference at this point in the manuscript:

  1. Schober, P.; Boer, C.; Schwarte, L.A. Correlation Coefficients: Appropriate Use and Interpretation. Anesth Analg 2018, 126, 1763–1768, doi:10.1213/ANE.0000000000002864.

Comment 6: "Page 4 line 144, 'a specified mass error tolerance (ppm)', what is the ppm value that was used?" 

Response:  Thank you for raising this important question. We appreciate the opportunity to clarify the methodology for isotopic peak identification in SMQVP.

The "specified mass error tolerance (ppm)" on Page 4, line 144 is a user-defined parameter in SMQVP. This allows researchers to set the mass tolerance based on the mass accuracy of their Mass spectrometer.

To improve clarity in the manuscript, we have revised the sentence to indicate that this is a user-specified parameter and briefly explain its function in matching theoretical isotopic mass differences: “SMQVP utilizes the isotopologues function from MetaboCoreUtils [20] for preliminary identification of potential isotopic peak pairs. This involves searching for peak pairs whose measured mass difference deviates from the theoretical isotopic mass difference (e.g., ~1.00335 Da for the M+1 isotope) by no more than a user-specified mass error tolerance (ppm).” (Page 4, Line 156)

Comment 7: "Page 4 line 168, 'The tool is accessible online at https://metax.genomics.cn/app/SMQVP, featuring a clear layout with a navigation panel on the left and an analysis content panel on the right (Figure 1B).' I did not see the analysis content panel on the right on the website." 

Response:  Thank you for pointing out this potential ambiguity. To clarify, the analysis content panel is indeed present on the right side of the SMQVP interface. We acknowledge that the original Figure 1B may not have sufficiently highlighted its layout.

We have revised Figure 1B to explicitly annotate both the navigation panel (left) and analysis content panel (right) using dashed boxes and labels (as shown in the updated figure).

We have double-checked the live tool at https://metax.genomics.cn/app/SMQVP and confirmed that the interface matches the annotated Figure 1B. Users can access the analysis panel by selecting any module (e.g., "Data Upload" or "Noise Filtering") from the navigation panel.

We appreciate your diligence in ensuring the accuracy of our descriptions and hope the revised figure resolves the confusion.

Comment 8: "Page line 179, 'These files include the processed peak intensity table, along with comprehensive tables containing detailed information for each peak (including intensity, missing ratio, isotope and adduct status, and noise score) and for each pixel (including intensity, missing ratio).' Missing ratio was not defined." 

Response: Thank you for highlighting this oversight. We acknowledge that the term "missing ratio" was not explicitly defined in the original manuscript.

There are two missing ratios, ion missing ratio and pixel missing ratio. To address this, we have revised the text to clarify its definition in Section 2.2.6 ("QVP6: Assessment of Missing Value Distribution") as following: "QVP6 evaluates the distribution of missing values by calculating two metrics: (1) the pixel missing ratio, defined as the percentage of ions undetected in a given pixel, and (2) the ion missing ratio, defined as the percentage of pixels where a specific ion was undetected. Spatial visualization of these ratios identifies regions or ions with poor data coverage. " (Page 4, Line 147)

To align with these definitions, we have updated the description in line 179: "These files include the processed peak intensity table, along with comprehensive tables containing detailed information for each peak (including ion intensity, ion missing ratio, isotope and adduct status, and noise score) and for each pixel (including pixel intensity and pixel missing ratio)." (Page 4, Line 200)

Comment 9: "Page 6 line 247, reference 'Louvain clustering[17]' did not use this term. Need clarify." 

Response: Thank you for highlighting this citation concern. We appreciate the opportunity to clarify our rationale for the original reference and ensure methodological transparency.

We initially cited the Seurat v5 reference as a general acknowledgment of the Seurat toolkit’s broader utility in single-cell analysis. While Louvain clustering is a core Seurat function, we inadvertently referenced the latest Seurat publication instead of the foundational paper describing its implementation.

To ensure accurate attribution, we have added the following reference: Stuart, T.; Butler, A.; Hoffman, P.; et al. Comprehensive Integration of Single-Cell Data. Cell 2019, 177, 1888–1902.

This paper formally describes the Louvain clustering workflow in Seurat, making it the appropriate citation for methodological reproducibility. The revised sentences is: " To demonstrate the impact of SMQVP's preprocessing steps on downstream analysis, we performed Louvain clustering [23,24] on the data both before and after processing." (Page 7, Line 286)

We sincerely apologize for the initial oversight and appreciate your meticulous review, which has strengthened the manuscript’s scholarly rigor.

Comment 10: "Figure 2 is not clear. Labels are not readable, so I can’t review Figure 2." 

Response: Thank you for highlighting the clarity issues with Figure 2, specifically the unreadable labels. We agree that clear and legible figures are essential for effective communication and apologize that the initial submission of Figure 2 did not meet these standards, hindering your ability to thoroughly assess it.

To address these concerns, we have thoroughly revised Figure 2. We've increased the font sizes of all labels to ensure readability, added clear subtitles to each subfigure for enhanced understanding, and optimized the layout of certain subfigures to improve overall clarity and prevent confusion. Additionally, we have also revised the figure caption to better describe the content of each panel. 

Revised caption: “Figure 2. Quality visualization results of SMQVP. Hematoxylin and Eosin (H&E) staining image (A); QVP1: Mass spectra analysis of background and tissue areas (B); QVP2: Background Similarity shown by a correlation heatmap of mass spectra from background areas (C); QVP3: Proportion of tissue enriched ions (D); QVP4: Distribution of noise scores (E); QVP5: Intensity distribution of pixels (F) and ions (G); QVP6: Missing value distribution of pixels (H) and ions (I); QVP7: Distribution of isotopic peaks (J) and QVP8: Distribution of adduct ions (K).” (Page 8, Line 300)

We sincerely appreciate your detailed review and constructive comments, which have significantly improved the manuscript.

Reviewer 2 Report

Comments and Suggestions for Authors

The authors developed a web application specializing in Quality Visualization and Processing of Mass Spectrometry Imaging data. This app should have great potential to make MS imaging data preprocessing more convenient for general users. This work should gain a broad interest in the mass spectrometry and spatial metabolomics community. It can be considered for publication after the reviewers’ comments below can be well addressed.

  1. Line 33-35, conventionally, global researchers in the MS imaging community consider MALDI, DESI, and SIMS as the three representative techniques. AFADESI may outperform DESI in sensitivity and a wide molecular coverage, but it is still derived from DESI.
  2. Line 35-37, to the reviewer’s understanding, the central topic is about developing a software that is applicable to all types of in situ ionization for MS imaging purposes. So, the statement here about performance comparison between MALDI and DESI/AFADESI is not necessary. Furthermore, this statement is not unbiased either because MALDI should also be applied for spatial metabolomics such as Chem.2025, 97, 14, 7986–7994.
  3. Line 37, “all mass spectrometry imaging techniques inherent challenges.” This is not a sentence.
  4. Please specify what types of data format (e.g. mzXML, CDF) that the developed software is compatible with.
  5. Line 43-46, there have been a lot of software, packages, or platforms that have functions of data preprocessing summarized in the literature (Table 2, Acta Materia Medica, 2022, Volume 1, Issue 4, p. 507-533). Can authors make a brief introduction to highlight the feature of SMQVP compared to these existent ones?
  6. Line 71-73, similar strategy of the noise ion filtering stated here has been proposed by the reference: Chem.2023, 95, 17, 6775–6784, which should be cited.
  7. Figure 2, critical values and annotations are difficult for readers to observe. For example, line 219-line 223, the authors stated that 37% of the detected ions were enriched in the tissue, where does this percentage come from in Figure 2D and supplementary Figure 3? In addition, should this threshold be determined by the users’ personal determination or based on a certain objective metric?
  8. For potential users’ convenience, please provide more details about the data format and maximum data size (MB, GB, TB) that this software can address

Minor:

  1. The abbreviation of air flow-assisted desorption electrospray ionization should be “AFADESI” other than “AFAD-ESI”.
  2. Line 46, change “.” to “,”
  3. Typo: Figure 2 caption (Line 264), “visualizayion”
Comments on the Quality of English Language

A thorough proofing is strongly suggested.

Author Response

1.     Summary

We sincerely appreciate your valuable comments, which have significantly improved the technical standardization and usability of SMQVP. The revised manuscript now clearly identifies MALDI, DESI, AFADESI, and SIMS as the core technical systems for mass spectrometry imaging, eliminating inappropriate descriptions of platform performance and standardizing technical terms such as AFADESI. By adding a feature comparison table (Supplementary Table 1) and conducting comparative analyses with commonly used tools, we have further strengthened the manuscript. In terms of graphical presentation, we have included screenshots for the proportion of tissue-enriched ions (37%). These enhancements better demonstrate the scientific value and cross-platform applicability of SMQVP as a quality control solution for spatial metabolomics.

2. Point-by-point response to Comments and Suggestions for Authors

Comment 1: Line 33-35, conventionally, global researchers in the MS imaging community consider MALDI, DESI, and SIMS as the three representative techniques. AFADESI may outperform DESI in sensitivity and a wide molecular coverage, but it is still derived from DESI.

Response 1: Thank you for this insightful comment. We fully agree with your perspective that MALDI, DESI, and SIMS are globally recognized as the three foundational technologies in mass spectrometry imaging (MSI). AFADESI, while demonstrating enhanced sensitivity and broader molecular coverage compared to DESI, is indeed an advanced derivative of DESI. We have revised the text to align with this classification and added the reference to contextualize SIMS’s role in MSI.

Revised Text: “This field is supported by several key technological platforms, including Matrix-Assisted Laser Desorption/Ionization Mass Spectrometry Imaging (MALDI-MSI) [4,5], Desorption Electrospray Ionization Mass Spectrometry Imaging (DESI-MSI) [6], air flow-assisted de-sorption electrospray ionization (AFADESI) [7] and Secondary Ion Mass Spectrometry (SIMS) [8].” (Page 1, Line 37).

Reference Updates: 8. Anderton, C.R.; Gamble, L.J. Secondary Ion Mass Spectrometry Imaging of Tissues, Cells, and Microbial Systems. Micros Today 2016, 24, 24–31, doi:10.1017/s1551929516000018.

Thank you for guiding us toward a more accurate representation of MSI technologies.

Comment 2: Line 35-37, to the reviewer’s understanding, the central topic is about developing a software that is applicable to all types of in situ ionization for MS imaging purposes. So, the statement here about performance comparison between MALDI and DESI/AFADESI is not necessary. Furthermore, this statement is not unbiased either because MALDI should also be applied for spatial metabolomics such as Chem.2025, 97, 14, 7986–7994.

Response 2: Thank you for your critical observation and suggestion. We fully agree that the original statement comparing MALDI and DESI/AFADESI performance was unnecessary and potentially biased. To address this, we have removed the comparison and revised the text to focus on the broad applicability of SMQVP across MSI platforms. Additionally, we have also added the suggested reference to emphasize MALDI’s established role in spatial metabolomics.

Reference Updates: 4.   Zhang, Y.; Chen, P.; Geng, H.; Li, M.; Chen, S.; Ma, B.; Ma, Y.; Lai, J.; Cui, X.; Chong, W.; et al. Development of a Single-Cell Spatial Metabolomics Method for the Characterization of Cell–Cell Metabolic Interactions. Anal Chem 2025, 97, 7986–7994, doi:10.1021/acs.analchem.5c00384.

Comment 3: Line 37, “all mass spectrometry imaging techniques inherent challenges.” This is not a sentence.

Response 3: Thank you for highlighting this grammatical error. We sincerely apologize for the oversight and have revised the sentence to ensure clarity and grammatical correctness. The original sentence has been corrected as follows: "All mass spectrometry imaging techniques have inherent challenges." (Page 2, Line 42)

Comment 4: Please specify what types of data format (e.g. mzXML, CDF) that the developed software is compatible with.

Response 4: Thank you for your query regarding data format compatibility. To clarify, SMQVP is designed to accept standardized peak table matrices as input, ensuring flexibility across spatial metabolomics platforms. The required format is structured as follows: “SMQVP requires input data in the form of a peak table matrix, where the first two columns represent the X and Y spatial coordinates of each pixel, followed by columns for different m/z values (ions), with each cell indicating the ion intensity at that pixel.” (Page 5, Line 192).

While SMQVP does not directly process raw mass files, it is compatible with data from any MSI platform once converted to the peak table format.

We have revised the manuscript to explicitly state this compatibility (Section 2.3): “This peak table format makes SMQVP compatible with data generated by various spatial mass spectrometry imaging platforms, such as MALDI-MSI, DESI-MSI, and AFADESI, upon conversion of raw data (raw, CDF, imzML) to this format.” (Page 5, Line 195)

Thank you for prompting this clarification, which enhances the manuscript’s technical transparency.

Comment 5: Line 43-46, there have been a lot of software, packages, or platforms that have functions of data preprocessing summarized in the literature (Table 2, Acta Materia Medica, 2022, Volume 1, Issue 4, p. 507-533). Can authors make a brief introduction to highlight the feature of SMQVP compared to these existent ones?

Response 5: Thank you for highlighting the importance of contextualizing SMQVP’s capabilities within the landscape of existing spatial metabolomics software. We have revised the introduction to explicitly compare SMQVP’s features against widely used tools (e.g., Cardinal, MassImager, multi-MSIProcessor) and added the suggested reference to Acta Materia Medica (2022) for broader context.

To highlight SMQVP's feature in relation to existing software, we have performed a qualitative feature comparison. This comparison is presented in the manuscript as Supplementary Table 2.

Feature

SMQVP

Cardinal [1]

MSIReader [2]

MassImager [3]

multi-MSIProcessor [4]

Background Region Analysis

✔️ Spectral comparison, Pearson correlation (QVP1, QVP2)

Tissue-Enriched Ion Identification

✔️ Wilcoxon rank-sum test, fold-change (QVP3)

Noise Ion Filtering

✔️ Quadrat test for spatial randomness (QVP4)

Ion intensity and missing Value Assessment

✔️ Pixel/ion missing ratios, spatial visualization (QVP6)

✔️

✔️

✔️

✔️

Isotopic Peak Identification

✔️ MetaboCoreUtils + spatial correlation (QVP7)

✔️

✔️

Adduct Ion Analysis

✔️ Customizable adduct lists + spatial correlation (QVP8)

✔️

Interactive Visualization

✔️ Shiny-based GUI with real-time adjustments

✔️

✔️

✔️

Open Source

✔️ GitHub

✔️

✔️ 

References:

  1. Bemis, K.D.; Harry, A.; Eberlin, L.S.; Ferreira, C.; van de Ven, S.M.; Mallick, P.; Stolowitz, M.; Vitek, O. Cardinal: An R Package for Statistical Analysis of Mass Spectrometry-Based Imaging Experiments. Bioinformatics 2015, 31, 2418–2420, doi:10.1093/bioinformatics/btv146.
  2. Bokhart, M.T.; Nazari, M.; Garrard, K.P.; Muddiman, D.C. MSiReader v1.0: Evolving Open-Source Mass Spectrometry Imaging Software for Targeted and Untargeted Analyses. J Am Soc Mass Spectrom 2018, 29, 8–16, doi:10.1007/s13361-017-1809-6.
  3. He, J.; Huang, L.; Tian, R.; Li, T.; Sun, C.; Song, X.; Lv, Y.; Luo, Z.; Li, X.; Abliz, Z. MassImager: A Software for Interactive and in-Depth Analysis of Mass Spectrometry Imaging Data. Anal Chim Acta 2018, 1015, 50–57, doi:https://doi.org/10.1016/j.aca.2018.02.030.
  4. Bi, S.; Wang, M.; Pu, Q.; Yang, J.; Jiang, N.; Zhao, X.; Qiu, S.; Liu, R.; Xu, R.; Li, X.; et al. Multi-MSIProcessor: Data Visualizing and Analysis Software for Spatial Metabolomics Research. Anal Chem 2024, 96, 339–346, doi:10.1021/acs.analchem.3c04192.

Here is the revised Introduction section with a sentence added to reference Supplementary Table 2: “A comparison of SMQVP's key features and capabilities relative to existing spatial metabolomics software is provided in Supplementary Table 2. (Page 2, Line 67)”

We have also included this reference in our paper to support our introduction: " Although several software tools exist for spatial metabolomics data analysis [12], such as commercial options like MassImager [13], and non-commercial packages like Cardinal [14], multi-MSIProcessor [15] and mzMINE3[16]. they often lack dedicated, comprehensive modules specifically for data quality assessment and systematic preprocessing." (Page 2, Line 48).

Here, [12] corresponds to the added citation: “Song, X.; Li, C.; Meng, Y. Mass Spectrometry Imaging Advances and Application in Pharmaceutical Research. Acta Materia Medica 2022, 1, 507–533.”

We appreciate your suggestion to strengthen this comparison and hope the revisions clarify SMQVP’s novel contributions to the field.

Comment 6: Line 71-73, similar strategy of the noise ion filtering stated here has been proposed by the reference: Chem.2023, 95, 17, 6775–6784, which should be cited.

Response 6: Thank you for directing us to this relevant reference (Chen et al., 2023, Anal. Chem., 95, 6775–6784). We agree that the strategy for noise ion filtering based on spatial randomness aligns closely with the methodology described in this work. To acknowledge this contribution, we have cited the reference in the revised manuscript. The updated text now reads: "Subsequently, noise ions, characterized by random spatial distributions [17], are identified and filtered out." (Page 2, Line 85).

Revised references: “17.       Song, X.; Zang, Q.; Zhang, J.; Gao, S.; Zheng, K.; Li, Y.; Abliz, Z.; He, J. Metabolic Perturbation Score-Based Mass Spectrometry Imaging Spatially Resolves a Functional Metabolic Response. Anal Chem 2023, 95, 6775–6784, doi:10.1021/acs.analchem.2c01723.”

We appreciate your guidance in strengthening the manuscript’s scholarly rigor.

Comment 7: Figure 2, critical values and annotations are difficult for readers to observe. For example, line 219-line 223, the authors stated that 37% of the detected ions were enriched in the tissue, where does this percentage come from in Figure 2D and supplementary Figure 3? In addition, should this threshold be determined by the users’ personal determination or based on a certain objective metric?

Response 7: Thank you for raising these critical points regarding the clarity of the tissue-enriched ion proportion and threshold determination. We acknowledge that the original 37% value was not directly visible in Figure 2 or Supplementary Figure 3 and have revised the manuscript and supplementary materials to address this. We have added a new subfigure (B) to Supplementary Figure 3, which now includes a screenshot of SMQVP’s interface explicitly displaying the calculated proportion (37%) of tissue-enriched ions.

As suggested by Reviewer 2, we replaced the t-test with the Wilcoxon rank-sum test for identifying tissue-enriched ions. We added the following statement to clarify variability in tissue-enriched ion proportions: "The proportion of tissue-enriched ions varies across experimental platforms (e.g., MALDI vs. DESI) and tissue types. While no universal threshold exists, higher proportions generally indicate superior data quality by reflecting stronger retention of tissue-specific signals." (Page 6, Line 252)

These revisions provide actionable guidance for threshold customization. We appreciate your feedback, which has strengthened the manuscript’s reproducibility and user-friendliness.

Comment 8: For potential users’ convenience, please provide more details about the data format and maximum data size (MB, GB, TB) that this software can address

Response 8: Thank you for your suggestion to clarify SMQVP’s data format compatibility and processing scalability. We have added a description to the data data format: “SMQVP requires input data in the form of a peak table matrix, where the first two columns represent the X and Y spatial coordinates of each pixel, followed by columns for different m/z values (ions), with each cell indicating the ion intensity at that pixel. This peak table format makes SMQVP compatible with data generated by various spatial mass spec-trometry imaging platforms, such as MALDI-MSI, DESI-MSI, and AFADESI, upon conversion of raw data to this format.” (Page 5, Line 192).

To address the processing scalability, we have added explicit guidance on performance benchmarks in the revised manuscript. The online version of SMQVP is currently hosted on a cloud server equipped with a 128 CPU cores and 1000 GB of RAM, which is provisioned to adequately handle typical spatial metabolomics dataset sizes encountered in current research.

The memory consumption and runtime required for SMQVP to process these datasets are detailed in the newly added Supplementary Table 3:

Dataset

Mouse Brain (Low-res)

Mouse Brain (Demo)

Human Stomach (High-res)

Spatial Resolution/μm

100

50

20

Pixel Numbers

21,164

53,812

118,604

Ion Numbers

2,843

2,654

2,898

Raw data size/GB

8.8

20.8

77

Peak table size/MB

298

714.3

1700

Average Memory Usage/GB

4.27

5.1

10.47

Runtime/min

3

6

25

These results demonstrate SMQVP’s capability to handle datasets as large as 1.7GB. We have integrated a detailed performance evaluation into the manuscript and revised Section 2.3 as follows: " The tool is accessible online at https://metax.genomics.cn/app/SMQVP, hosted on a cloud server equipped with 128 CPU cores and 1000 GB of RAM (Page 4, Line 183)"; and “Details on SMQVP's computational performance, including execution time and memory usage on various dataset sizes, are provided in Supplementary Table 3” (Page 5, Line 188).

While the online platform is well-suited for small or median dataset sizes, we acknowledge that processing ultra-large datasets that significantly exceed these scales might encounter practical limitations related to web server capacity or data transfer speeds. For researchers working with such ultra-large datasets, we recommend leveraging the provided source code for local deployment. This allows them to utilize dedicated high-performance computing resources that can be scaled to handle the specific volume and complexity of their data.

Comment 9: The abbreviation of air flow-assisted desorption electrospray ionization should be “AFADESI” other than “AFAD-ESI”.

Response 9: Thank you for pointing out the inconsistency in the abbreviation of air flow-assisted desorption electrospray ionization. We fully agree with adopting the standardized abbreviation "AFADESI" and have revised all instances of "AFAD-ESI" to "AFADESI" throughout the manuscript.

Your meticulous attention to detail has significantly improved the clarity and consistency of the manuscript. Thank you for this valuable correction.

Comment 10: Line 46, change “.” to “,”

Response 10: Thank you for identifying the punctuation error in Line 46. We have revised the sentence to replace the erroneous period with a comma, ensuring grammatical correctness and proper sentence structure. The updated text now reads:"Although several software tools exist for spatial metabolomics data analysis, such as commercial options like MassImager [9], and non-commercial packages like Cardinal [10], multi-MSIProcessor [11] and mzMINE3 [12], they often lack dedicated, comprehensive modules specifically for data quality assessment and systematic preprocessing." (Page 2, Line 48).

This revision eliminates the fragment and improves readability. We appreciate your meticulous review, which has enhanced the clarity of our manuscript.

Comment 11: Typo: Figure 2 caption (Line 264), “visualizayion”

Response 11: Response: Thank you for catching this typographical error. We sincerely apologize for the oversight and have revised the Figure 2 caption to correct "visualizayion" to "visualization". The updated caption now reads:

" Figure 2. Quality visualization results of SMQVP. Hematoxylin and Eosin (H&E) staining image (A); QVP1: Mass spectra analysis of background and tissue areas (B); QVP2: Background Simi-larity shown by a correlation heatmap of mass spectra from background areas (C); QVP3: Pro-portion of tissue enriched ions (D); QVP4: Distribution of noise scores (E); QVP5: Intensity distri-bution of pixels (F) and ions (G); QVP6: Missing value distribution of pixels (H) and ions (I); QVP7: Distribution of isotopic peaks (J) and QVP8: Distribution of adduct ions (K)." (Page 8, Line 300)

Additionally, we have performed a full-text spelling and grammar check to ensure no similar errors persist in the manuscript.

Thank you for your meticulous review and for helping us improve the clarity of the manuscript.

Reviewer 3 Report

Comments and Suggestions for Authors

The manuscript introduces SMQVP, a web application for special metabolomics data quality control, addressing a critical gap in current tools. While the tool looks promising, several improvements can strengthen the paper-

  • While the tool is novel, the manuscript would be significantly stronger if it quantitatively compared SMQVP’s performance against tools like Cardinal, MSIReader, or multi-MSIProcessor. This would situate SMQVP's value in a measurable context.
  • In Introduction section, briefly comment on how SMQVP behaves with MALDI or DESI data. Even one demo dataset would strengthen generalizability claims.
  • I would also suggest to include a schematic diagram showing how each QVP contributes to the workflow, to support users new to metabolomics/MSI.
  • Several QVPs involve user-defined thresholds (e.g., QVP4 noise score, QVP3 fold-change cutoff). The manuscript lacks guidance or automated support for optimal threshold selection, which may lead to inconsistent use.
  • The paper does not address the computational performance of SMQVP for larger datasets (>100,000 pixels or >10,000 ions). This is a key concern for practical deployment in high-throughput studies.
  • Line 31, support the statement with 1016/j.ebiom.2023.104627
  • The t-test used for tissue enriched ion identification assumes normality, which may not hold for metabolomics data. Non- parametric alternatives for the t-test’s applicability should be used.
  • QVP4 may oversimplify spatial complexity. Comparing it to other spatial autocorrelation methods would validate its effectiveness.
  • Correlation based identification risks false positives. Incorporating MS/MS validation or known standards would strengthen claims.
  • Only AFAD-ESI data is tested. There’s no indication whether the tool generalizes to MALDI, DESI, or other MSI platforms — even a brief test or acknowledgment would enhance credibility.
  • Some terms are dense (e.g., “tissue-enriched ions via total intensity fold thresholding”) without visual or conceptual aid. A few visual metaphors or simplified descriptions would help interdisciplinary readers.
  • If possible, I would suggest to Provide Threshold Recommendations or Auto-tuning. Either recommend empirical default values for thresholds (like a noise score of 90) or introduce an optional “auto mode” where the tool proposes thresholds based on statistical distributions.
  • Add a performance analysis section with SMQVP’s execution time and memory usage on various dataset sizes. This is especially important for real-time GUI tools.
  • Typo in Figure 2 caption: “visualizayion” → should be “visualization”.
  • Avoid passive voice in technical descriptions: e.g., “Ions were identified…” → “SMQVP identifies ions…”.
  • Consider making Figure 1B more readable by enlarging font size in the screenshot.
  • The software URL should be hyperlinked properly in-text for ease of access.

Author Response

  1. Summary

We sincerely thank you for the constructive feedback, which has strengthened SMQVP’s methodological rigor. The revised manuscript now includes a qualitative feature comparison with existing tools (Supplementary Table 2), validates SMQVP across both AFADESI and MALDI platforms, and incorporates non-parametric statistical methods for robust tissue-enriched ion identification. We added performance benchmarks (Supplementary Table 3) demonstrating efficient handling of large datasets. These updates clarify SMQVP’s unique role in spatial metabolomics quality control while addressing computational scalability and cross-platform generalizability.

2. Point-by-point response to Comments and Suggestions for Authors

Comment 1: "While the tool is novel, the manuscript would be significantly stronger if it quantitatively compared SMQVP’s performance against tools like Cardinal, MSIReader, or multi-MSIProcessor. This would situate SMQVP's value in a measurable context." 

Response 1: Thank you for the constructive suggestion to compare SMQVP with existing tools. We agree that a comparison of capabilities is essential to demonstrate SMQVP's unique contribution.

To address this point and situate SMQVP's value in relation to existing software, we have performed a qualitative feature comparison. This comparison is presented in the manuscript as Supplementary Table 2.

Feature

SMQVP

Cardinal [1]

MSIReader [2]

MassImager [3]

multi-MSIProcessor [4]

Background Region Analysis

✔️ Spectral comparison, Pearson correlation (QVP1, QVP2)

Tissue-Enriched Ion Identification

✔️ Wilcoxon rank-sum test, fold-change (QVP3)

Noise Ion Filtering

✔️ Quadrat test for spatial randomness (QVP4)

Ion intensity and missing Value Assessment

✔️ Pixel/ion missing ratios, spatial visualization (QVP6)

✔️

✔️

✔️

✔️

Isotopic Peak Identification

✔️ MetaboCoreUtils + spatial correlation (QVP7)

✔️

✔️

Adduct Ion Analysis

✔️ Customizable adduct lists + spatial correlation (QVP8)

✔️

Interactive Visualization

✔️ Shiny-based GUI with real-time adjustments

✔️

✔️

✔️

Open Source

✔️ GitHub

✔️

✔️ 

References:

  1. Bemis, K.D.; Harry, A.; Eberlin, L.S.; Ferreira, C.; van de Ven, S.M.; Mallick, P.; Stolowitz, M.; Vitek, O. Cardinal: An R Package for Statistical Analysis of Mass Spectrometry-Based Imaging Experiments. Bioinformatics 2015, 31, 2418–2420, doi:10.1093/bioinformatics/btv146.
  2. Bokhart, M.T.; Nazari, M.; Garrard, K.P.; Muddiman, D.C. MSiReader v1.0: Evolving Open-Source Mass Spectrometry Imaging Software for Targeted and Untargeted Analyses. J Am Soc Mass Spectrom 2018, 29, 8–16, doi:10.1007/s13361-017-1809-6.
  3. He, J.; Huang, L.; Tian, R.; Li, T.; Sun, C.; Song, X.; Lv, Y.; Luo, Z.; Li, X.; Abliz, Z. MassImager: A Software for Interactive and in-Depth Analysis of Mass Spectrometry Imaging Data. Anal Chim Acta 2018, 1015, 50–57, doi:https://doi.org/10.1016/j.aca.2018.02.030.
  4. Bi, S.; Wang, M.; Pu, Q.; Yang, J.; Jiang, N.; Zhao, X.; Qiu, S.; Liu, R.; Xu, R.; Li, X.; et al. Multi-MSIProcessor: Data Visualizing and Analysis Software for Spatial Metabolomics Research. Anal Chem 2024, 96, 339–346, doi:10.1021/acs.analchem.3c04192.

And here is the revised Introduction section with a sentence added to reference Supplementary Table 2: “A comparison of SMQVP's key features and capabilities relative to existing spatial metabolomics software is provided in Supplementary Table 2. (Page 2, Line 67)”

Comment 2: "In Introduction section, briefly comment on how SMQVP behaves with MALDI or DESI data. Even one demo dataset would strengthen generalizability claims." 

Response 2: We appreciate your valuable feedback regarding the demonstration of SMQVP's compatibility with diverse spatial metabolomics platforms. To address this important point, we have strengthened the generalizability claims by validating SMQVP on both AFADESI and MALDI-MSI datasets, integrating the detailed results into the manuscript.

To strenghten this claim, we analyzed an AP-SMALDI dataset (OMIX010192). This dataset was processed through SMQVP’s full workflow, providing comprehensive evaluation results. The revised text within the manuscript reflecting this analysis states: “Representative analytical results derived from SMQVP's processing of this AP-SMALDI dataset are depicted in Supplementary Figures 7-14, thereby underscoring its consistent performance across diverse technological platforms.” (Page 5, Line 221)

Specifically, the Introduction section has been updated to include explicit compatibility statements and illustrative dataset examples, now reading: “SMQVP is platform-agnostic and compatible with data from diverse spatial metabolomics technologies. In this study, we have validated SMQVP’s performance using both AFAD-ESI-based dataset and MALDI-MSI datasets.” (Page 2, Line 65)

These revisions collectively underscore SMQVP’s versatility and provide researchers with clear examples of its applicability across MALDI, DESI, and other MSI technologies. We are grateful for your insightful comments, which have significantly enhanced the clarity and impact of our manuscript.

Comment 3: "I would also suggest to include a schematic diagram showing how each QVP contributes to the workflow, to support users new to metabolomics/MSI." 

Response 3: Thank you for your constructive suggestion regarding a schematic diagram to illustrate the contribution of each QVP to the workflow. We agree that clarifying these contributions is crucial for users new to metabolomics/MSI.

To address this valuable point, we have developed and included a categorization table (Supplementary Table 1) that systematically outlines the role of each QVPs in assessing data quality across key dimensions. This table organizes QVPs into three primary categories: Contamination Detection, Overall Data Quality, and Ion Credibility, with specific subcategories detailing their respective foci (e.g., "Signal Stability," "Proportion of Noise Ions"). For instance, QVP1 evaluates contaminant peaks through spectral comparisons between background and tissue regions, while QVP7 assesses isotopic peak ratios to validate ion reliability. This structured framework offers an intuitive guide to SMQVP’s quality control logic, significantly improving accessibility for novice researchers.

Furthermore, we have revised the relevant text within the manuscript to complement this table and further clarify the QVP functionalities: “SMQVP offers a robust suite of tools, encompassing eight quality visualization points (QVPs) and three core preprocessing functions, including background pixel removal, noise ion filtering, and identification of isotopic and adduct ions. Its visualization modules enable comprehensive assessments, such as evaluating specific ion spatial distributions, analyzing background region consistency, examining overall intensity and missing value distributions, and assessing isotopic peak and adduct ion ratios (Supplementary Table 1).” (Page 2, Line 59)

Supplementary Table 1: Data quality dimensions and their corresponding Quality Visualization Points (QVPs)

Category

Subcategory

Corresponding QVP

Contamination Detection

Contaminant Peak Detection (e.g., polymers)

QVP1: Spectral Analysis of Background and Tissue Regions

Overall Data Quality

Signal Stability during Data Acquisition

QVP2: Background Region Consistency

Overall Signal Intensity

QVP5: Analysis of Intensity Distribution

Overall Missing Rate

QVP6: Assessment of Missing Value Distribution

Ion Credibility

High Ion Intensity in Tissue Regions

QVP3: Proportion of Tissue-Enriched Ions

Proportion of Noise Ions

QVP4: Identification and Proportion of Noise Ions

Proportion of Ions with Isotopic Peaks

QVP7: Isotopic Peak Ratio Assessment

Proportion of Ions for which Adduct Forms Can Be Calculated

QVP8: Adduct Ion Ratio Assessment

We are grateful for your insightful feedback, which has undoubtedly strengthened the clarity and comprehensiveness of our manuscript.

Comment 4: "Several QVPs involve user-defined thresholds (e.g., QVP4 noise score, QVP3 fold-change cutoff). The manuscript lacks guidance or automated support for optimal threshold selection, which may lead to inconsistent use." 

Response 4: Thank you for highlighting the importance of guiding users in threshold selection for SMQVP’s quality control workflows.

We understand the concern that relying on user-defined thresholds without guidance or automated support may not be straightforward for some user. However, we know that the optimal threshold for many of these metrics can vary significantly across different datasets, depending on factors such as the MSI platform used, acquisition parameters, and the complexity of the biological sample. For this reason, a simple automated method may not be universally applicable.

Instead, SMQVP provides essential guidance through default values for all adjustable parameters, offering users a reasonable starting point. More importantly, it incorporates dedicated visualizations specifically designed to empower users to make informed, data-specific decisions about these thresholds. For example, as described in Section 2.2.4, SMQVP generates spatial distribution maps for ions near the noise threshold, enabling users to visually assess and optimize the noise score threshold based on their dataset's characteristics.

To clarify these points, we have updated Section 2.2.4: "SMQVP provides empirically derived default thresholds (e.g., noise score = 60) but encourages users to iteratively refine these values using interactive visualizations. To aid in threshold selection, SMQVP generates spatial distribution maps for ions near the threshold. This functionality allows users to visually assess whether these borderline ions exhibit a random spatial distribution, thereby facilitating the precise optimization of the noise score threshold based on their dataset's characteristics." (Page 3, Line 133)

Furthermore, this process of visually exploring the data under different parameter settings and evaluating the impact of threshold choices not only facilitates robust threshold selection but also helps researchers gain a deeper understanding of their specific dataset's spatial characteristics and quality issues.

By combining empirical defaults, interactive visualization, and user-driven adjustments, SMQVP balances flexibility and reliability across diverse spatial metabolomics studies. We thank the reviewer for underscoring the importance of threshold guidance and remain committed to enhancing these features in future updates.

Comment 5: "The paper does not address the computational performance of SMQVP for larger datasets (>100,000 pixels or >10,000 ions). This is a key concern for practical deployment in high-throughput studies." 

Response: Thank you for emphasizing the importance of computational performance for high-throughput spatial metabolomics studies. We agree that demonstrating performance on varying dataset sizes is a key concern for the practical utility and deployment of spatial metabolomics analysis tools.

The online version of SMQVP is currently hosted on a cloud server equipped with a 128 CPU cores and 1000 GB of RAM, which is provisioned to adequately handle typical spatial metabolomics dataset sizes encountered in current research.

The memory consumption and runtime required for SMQVP to process these datasets are detailed in the newly added Supplementary Table 3:

Dataset

Mouse Brain (Low-res)

Mouse Brain (Demo)

Human Stomach (High-res)

Spatial Resolution/μm

100

50

20

Pixel Number

21,164

53,812

118,604

Ion Number

2,843

2,654

2,898

Raw data size/GB

8.8

20.8

77

Peak table size/MB

298

714.3

1700

Average Memory Usage/GB

4.27

5.1

10.47

Runtime/min

3

6

25

These results demonstrate SMQVP’s capability to handle datasets with >100,000 pixels and >3,000 ions within practical timeframes using the current cloud infrastructure. We have integrated a detailed performance evaluation into the manuscript and revised Section 2.3 as follows: " The tool is accessible online at https://metax.genomics.cn/app/SMQVP, hosted on a cloud server equipped with 128 CPU cores and 1000 GB of RAM. (Page 4, Line 183)", and “Details on SMQVP's computational performance, including execution time and memory usage on various dataset sizes, are provided in Supplementary Table 3.” (Page 5, Line 188).

While the online platform is well-suited for common dataset sizes, we acknowledge that processing ultra-large datasets that significantly exceed these scales might encounter practical limitations related to web server capacity or data transfer speeds. For researchers working with such ultra-large datasets, we recommend leveraging the provided source code for local deployment. This allows them to utilize dedicated high-performance computing resources that can be scaled to handle the specific volume and complexity of their data.

Comment 6: "Line 31, support the statement with 1016/j.ebiom.2023.104627." 

Response: Thank you for your suggestion to incorporate the referenced study (10.1016/j.ebiom.2023.104627) into the manuscript. We have revised the introduction to better contextualize the significance of metabolomics in biomedical research and included this citation as reference [3]. Below are the details of the revision:

"Metabolomics is a rapidly advancing field of considerable significance in modern biology and biomedical research [1–3]. Among the various metabolomic approaches, spatial metabolomics is particularly powerful. By enabling the mapping of metabolite distributions with spatial resolution at the tissue and even cellular levels, it offers powerful capabilities for elucidating biological processes and disease mechanisms. This field is supported by several key technological platforms, including Matrix-Assisted Laser De-sorption/Ionization Mass Spectrometry Imaging (MALDI-MSI) [4,5], Desorption Elec-trospray Ionization Mass Spectrometry Imaging (DESI-MSI) [6], air flow-assisted desorp-tion electrospray ionization (AFADESI) [7] and Secondary Ion Mass Spectrometry (SIMS) [8]. all mass spectrometry imaging techniques have inherent challenges [9][10]." (Page 1, Line 33)

This revision strengthens the manuscript by linking spatial metabolomics to the broader metabolomics field and highlighting its translational potential. We appreciate your guidance in enhancing the context of our work.

Comment 7: "The t-test used for tissue enriched ion identification assumes normality, which may not hold for metabolomics data. Non-parametric alternatives for the t-test’s applicability should be used." 

Response: Thank you for this insightful suggestion regarding the use of the t-test for identifying tissue-enriched ions. We agree with your concern that metabolomics data, depending on the specific ions and experimental conditions, may not always fully satisfy the normality assumption required for the t-test.

We have replaced the t-test with a non-parametric alternative, the Wilcoxon rank-sum test, for comparing ion intensities between the tissue and background groups. This change has been implemented in the SMQVP software code.

The description of QVP3 in the manuscript (Section 2.2.3) has been updated to reflect the use of the Wilcoxon rank-sum test for calculating p-values and identifying tissue-enriched ions. Revised text: “For each ion, a Wilcoxon rank-sum test is conducted to compare the intensity distribution between the tissue and background groups.” (Page 3, Line 114)

Comment 8: "QVP4 may oversimplify spatial complexity. Comparing it to other spatial autocorrelation methods would validate its effectiveness." 

Response: Thank you for this comment regarding QVP4 and the suggestion to compare its effectiveness with other spatial autocorrelation methods.

The primary objective of QVP4 is to identify ions that exhibit a statistically random spatial distribution, which is a characteristic often associated with technical noise or non-biologically relevant signals in spatial metabolomics data.

For this specific purpose, SMQVP utilizes the quadrat test from the spatstat package. The quadrat test is a standard and well-established statistical method specifically designed to test the null hypothesis of Complete Spatial Randomness (CSR) for a point pattern.

We appreciate the suggestion to compare this approach to spatial autocorrelation methods. While measures of spatial autocorrelation (such as Moran's I) are powerful tools for detecting the presence of spatial structure. Both Moran's I and the quadrat test take spatial information into consideration. However, the quadrat test is more computationally efficient and directly tests against the hypothesis of spatial randomness itself. For the specific goal of identifying ions that lack significant spatial structure and appear randomly distributed (consistent with noise), a direct test for complete spatial randomness (CSR) like the quadrat test is a statistically appropriate method.

We are confident that the quadrat test, used in conjunction with the visual inspection provided by SMQVP, is effective for its intended role in identifying and filtering spatially random noise ions. We have added further clarification in the manuscript text regarding the specific statistical basis of the quadrat test as a test for CSR to elaborate on why this method is suitable for identifying random spatial patterns. Revised text: “To address this, the SMQVP software utilizes the quadrat test from the spatstat package [14], a statistical test for Complete Spatial Randomness (CSR), to identify ions with statistically random spatial distributions. (Page 3, Line 126)”

Comment 9: "Correlation based identification risks false positives. Incorporating MS/MS validation or known standards would strengthen claims." 

Response: Thank you for raising this important methodological consideration. We fully acknowledge the limitations of correlation-based identification and the value of MS/MS validation in reducing false positives. Below is our detailed rationale for the current design of SMQVP and proposed mitigations:

Spatial metabolomics datasets are primarily acquired in MS1 profiling mode due to the inherent challenges of performing MS/MS fragmentation across thousands of pixels (e.g., time constraints, ion suppression).

SMQVP's focus is on addressing the upstream challenges of quality control and preprocessing for MS1 spatial profiling data to produce a cleaner, more reliable peak list for subsequent analysis and identification efforts. Therefore, incorporating MS/MS validation or standard matching, which belongs to the downstream structural identification stage, is outside the current scope and intended function of this quality control tool.

Comment 10: "Only AFAD-ESI data is tested. There’s no indication whether the tool generalizes to MALDI, DESI, or other MSI platforms — even a brief test or acknowledgment would enhance credibility." 

Response: We appreciate your insightful comment regarding the need to demonstrate SMQVP's compatibility with diverse Mass Spectrometry Imaging (MSI) platforms. We fully agree that validating the tool across multiple technologies is crucial for establishing its broad applicability and credibility.

To address this, we have significantly expanded our demo case to include a MALDI-MSI dataset (AP-SMALDI dataset, OMIX010192) alongside the original AFADESI data. This comprehensive validation is now integrated into the manuscript.

In the revised manuscript, we explicitly highlight SMQVP’s platform-agnostic design, as evidenced by the following statement: “Representative analytical results derived from SMQVP's processing of this AP-SMALDI dataset are depicted in Supplementary Figures 7-14, thereby underscoring its consistent performance across diverse technological platforms.” (Page 5, Line 221)

Furthermore, to clarify the tool's broader applicability, we have added a sentence in Section 2.3 that explicitly states its compatibility with data from various MSI platforms based on its standardized input format: “This standardized peak table format makes SMQVP compatible with data generated by various spatial mass spectrometry imaging platforms, such as MALDI-MSI, DESI-MSI, and AFAD-ESI, upon conversion of raw data to this format.” (Page 5, Line 196)

These revisions provide clear demonstration of SMQVP’s generalizability and directly address the need for cross-platform validation. Your feedback has significantly strengthened the technical rigor and transparency of our manuscript.

Comment 11: "Some terms are dense (e.g., 'tissue-enriched ions via total intensity fold thresholding') without visual or conceptual aid. A few visual metaphors or simplified descriptions would help interdisciplinary readers." 

Response: Thank you for your feedback on improving accessibility for interdisciplinary readers. We acknowledge the need to clarify terminology and enhance conceptual explanations. Below is a detailed response to address your concern:

The phrase "tissue-enriched ions via total intensity fold thresholding" does not appear in our manuscript. We have carefully reviewed the text to ensure alignment between terminology and methodology.

To directly address this potential point of confusion and align with your suggestion for visual aids, we have updated Supplementary Figure 3. Supplementary Figure 3C shows the total intensity image constructed using the tissue-enriched ions, and Supplementary Figure 3D illustrates how the total intensity threshold is applied to this image to effectively classify and remove background pixels. We believe these new panels serve as clear visual aids for this specific process.

We appreciate your emphasis on interdisciplinary accessibility. The revised text and figures aim to balance technical rigor with conceptual clarity, ensuring SMQVP’s methodology is comprehensible to readers across fields.

Comment 12: "If possible, I would suggest to Provide Threshold Recommendations or Auto-tuning. Either recommend empirical default values for thresholds (like a noise score of 90) or introduce an optional 'auto mode' where the tool proposes thresholds based on statistical distributions." 

Response: Thank you for the constructive suggestion regarding providing threshold recommendations or auto-tuning features. We agree that guiding users on threshold selection is important. In SMQVP, we have set default values for all adjustable parameters to provide a starting point (e.g., the default noise score threshold is 60, although this was not explicitly stated in the original text).

However, based on our experience and the diverse nature of spatial metabolomics data acquired from different sample types, instruments, and experimental conditions, the optimal threshold for distinguishing technical noise from biological signal, or for other quality metrics, can vary considerably. A fixed recommended value or simple auto-tuning method might not be universally applicable and could lead to suboptimal results or misinterpretation in specific datasets.

Therefore, our current design emphasizes user-guided threshold selection, which is supported by comprehensive visualizations within the tool. We believe this approach empowers users to make an informed decision based on the specific characteristics of their data, ensuring more reliable and data-appropriate quality control.

While full auto-tuning presents technical challenges and potential limitations in generalizability, we acknowledge its potential value for user convenience. We will consider exploring methods for providing data-driven threshold recommendations or developing more sophisticated auto-tuning features as a future enhancement to SMQVP.

Comment 13: "Add a performance analysis section with SMQVP’s execution time and memory usage on various dataset sizes. This is especially important for real-time GUI tools." 

Response: Thank you for emphasizing the importance of performance analysis for real-time GUI tools. We agree that computational efficiency and scalability are critical for practical deployment in high-throughput spatial metabolomics studies. To address this, we have integrated a detailed performance evaluation into the manuscript and revised Section 2.3 as follows: “The tool is accessible online at https://metax.genomics.cn/app/SMQVP, hosted on a cloud server equipped with 128 CPU cores and 1000 GB of RAM”. 4, Line 183)"; and “Details on SMQVP's computational performance, including execution time and memory usage on various dataset sizes, are provided in Supplementary Table 3.” (Page 5, Line 188).

Supplementary Table 3 demonstrates SMQVP’s capability to process datasets with >100,000 pixels and >3,000 ions within practical timeframes:

Dataset

Mouse Brain (Low-res)

Mouse Brain (Demo)

Human Stomach (High-res)

Spatial Resolution/μm

100

50

20

Pixel Number

21,164

53,812

118,604

Ion Number

2,843

2,654

2,898

Raw data size/GB

8.8

20.8

77

Peak table size/MB

298

714.3

1700

Average Memory Usage/GB

4.27

5.1

10.47

Runtime/min

2

3

18

These revisions underscore SMQVP’s ability to handle diverse dataset scales while maintaining usability for real-time quality control. We appreciate your feedback and hope the added performance analysis addresses your concerns.

Comment 14: "Typo in Figure 2 caption: 'visualizayion' → should be 'visualization'." 

Response: Thank you for catching this typographical error. We sincerely apologize for the oversight and have revised the Figure 2 caption to correct "visualizayion" to "visualization". The updated caption now reads:

" Figure 2. Quality visualization results of SMQVP. Hematoxylin and Eosin (H&E) staining image (A); QVP1: Mass spectra analysis of background and tissue areas (B); QVP2: Background Simi-larity shown by a correlation heatmap of mass spectra from background areas (C); QVP3: Pro-portion of tissue enriched ions (D); QVP4: Distribution of noise scores (E); QVP5: Intensity distri-bution of pixels (F) and ions (G); QVP6: Missing value distribution of pixels (H) and ions (I); QVP7: Distribution of isotopic peaks (J) and QVP8: Distribution of adduct ions (K)." (Page 8, Line 300).

Additionally, we have performed a full-text spelling and grammar check to ensure no similar errors persist in the manuscript.

Comment 15: "Avoid passive voice in technical descriptions: e.g., 'Ions were identified…' → 'SMQVP identifies ions…'." 

Response: Thank you for your insightful suggestion to improve the clarity of our technical descriptions by avoiding passive voice. We fully agree that using active voice enhances readability and better emphasizes SMQVP’s role as the primary agent in data processing workflows. To address this, we systematically revised the manuscript to ensure consistent use of active voice throughout.

For example, in the original text, the sentence “These tissue-enriched ions were used to construct a total intensity image of the sample” was revised to “SMQVP utilizes these tissue-enriched ions to construct a total intensity image of the sample. (Page 6, Line 255)” Similarly, the statement “Overall, isotopic peaks were identified for 14.44% of the ions” now reads “Overall, SMQVP identified isotopic peaks in 14.44% of the ions. (Page 7, Line 277)” These changes explicitly attribute actions to the tool, clarifying its functionality for readers.

We extended this revision to other instances of passive voice in the manuscript. For instance, “Pixels classified as background were removed from the dataset” was updated to “SMQVP removes pixels classified as background from the dataset using a threshold-based method as detailed in Section 2.2.3. (Page 2, Line 84)”. Thank you again for this valuable feedback.

Comment 16: "Consider making Figure 1B more readable by enlarging font size in the screenshot." 

Response: Thank you for your suggestion. We have revised Figure 1B by capturing a new screenshot with significantly enlarged font sizes for improved readability. The updated figure now clearly displays the layout of the SMQVP interface. We have verified that all labels and text in the figure are legible at both on-screen and print resolutions.

Comment 17: 

"The software URL should be hyperlinked properly in-text for ease of access." 

Response: Thank you for raising this issue. We have added a functional hyperlink to the software URL (https://metax.genomics.cn/app/SMQVP) in the revised manuscript. The updated text now reads: “The tool is accessible online at https://metax.genomics.cn/app/SMQVP, hosted on a cloud server equipped with 128 CPU cores and 1000 GB of RAM. (Page 4, Line 183)”

We have verified the link’s functionality to ensure seamless access for readers. Thank you again for your meticulous review.

Round 2

Reviewer 2 Report

Comments and Suggestions for Authors

The reviewer appreciates the authors' efforts to respond to all comments and make changes. This work can be considered for acceptance as its current form.